# Classical Swine Fever Virus Biology, Clinicopathology, Diagnosis, Vaccines and a Meta-Analysis of Prevalence: A Review from the Indian Perspective

**DOI:** 10.3390/pathogens9060500

**Published:** 2020-06-22

**Authors:** Yashpal Singh Malik, Sudipta Bhat, O. R. Vinodh Kumar, Ajay Kumar Yadav, Shubhankar Sircar, Mohd Ikram Ansari, Dilip Kumar Sarma, Tridib Kumar Rajkhowa, Souvik Ghosh, Kuldeep Dhama

**Affiliations:** 1Division of Biological Standardization, ICAR-Indian Veterinary Research Institute, Izatnagar, Bareilly, Uttar Pradesh 243001, India; sudiptabhat1991@gmail.com (S.B.); shubhankar.sircar@gmail.com (S.S.); mohd.ikram.ansari@gmail.com (M.I.A.); 2Division of Epidemiology, ICAR-Indian Veterinary Research Institute, Izatnagar, Bareilly, Uttar Pradesh 243122, India; vinodhkumar.rajendran@gmail.com; 3Animal Health, ICAR-National Research Centre on Pig (ICAR-NRCP), Guwahati, Assam 781015, India; dr.ajayyadav07@gmail.com; 4Department of Veterinary Microbiology, Assam Agricultural University, Khanapara, Guwahati 781022, India; dksarma1956@gmail.com; 5College of Veterinary Sciences & Animal Husbandry, Central Agricultural University, Selesih, Aizawl, Mizoram 796001, India; tridibraj09@gmail.com; 6Department of Biomedical Sciences, One Health Center for Zoonoses and Tropical Veterinary Medicine, Basseterre, St. Kitts PO Box 334, West Indies; sghosh@rossvet.edu.kn; 7Division of Pathology, ICAR-Indian Veterinary Research Institute, Izatnagar, Bareilly, Uttar Pradesh 243122, India

**Keywords:** classical swine fever virus, CSFV, swine, genome, phylogeny, diversity, immunobiology, diagnosis, vaccines, publication bias, prevalence, meta-analysis, India

## Abstract

Classical swine fever (CSF) is an economically significant, multi-systemic, highly contagious viral disease of swine world over. The disease is notifiable to the World Organization for Animal Health (OIE) due to its enormous consequences on porcine health and the pig industry. In India, the pig population is 9.06 million and contributes around 1.7% of the total livestock population. The pig industry is not well organized and is mostly concentrated in the eastern and northeastern states of the country (~40% of the country’s population). Since the first suspected CSF outbreak in India during 1944, a large number of outbreaks have been reported across the country, and CSF has acquired an endemic status. As of date, there is a scarcity of comprehensive information on CSF from India. Therefore, in this review, we undertook a systematic review to compile and evaluate the prevalence and genetic diversity of the CSF virus situation in the porcine population from India, targeting particular virus genes sequence analysis, published reports on prevalence, pathology, and updates on indigenous diagnostics and vaccines. The CSF virus (CSFV) is genetically diverse, and at least three phylogenetic groups are circulating throughout the world. In India, though genotype 1.1 predominates, recently published reports point toward increasing evidence of co-circulation of sub-genotype 2.2 followed by 2.1. Sequence identities and phylogenetic analysis of Indian CSFV reveal high genetic divergence among circulating strains. In the meta-analysis random-effects model, the estimated overall CSF prevalence was 35.4%, encompassing data from both antigen and antibody tests, and region-wise sub-group analysis indicated variable incidence from 25% in the southern to nearly 40% in the central zone, eastern, and northeastern regions. A country-wide immunization approach, along with other control measures, has been implemented to reduce the disease incidence and eliminate the virus in time to come.

## 1. Introduction

Classical swine fever (CSF), also known as hog cholera or more commonly swine fever, is a systemic, extremely contagious, and notifiable disease of viral origin affecting domestic and wild pigs [1]. The causative agent, the classical swine fever virus (CSFV), belongs to the genus *Pestivirus* in the family *Flaviviridae*. This genus also contains Bovine viral diarrhea virus (BVDV-1, 2), Border disease virus, Bungowannah virus, and HoBi-like virus. The disease was first reported in the year 1833 from the state of Ohio in the United States, and thereafter, the virus was identified in 1904 [2]. The incubation period in CSF disease varies between 3 and 10 days, and the course of the disease is influenced by the virulence of the virus and the age of the animals. CSF is characterized by several clinicopathological signs such as high fever (>40 °C), conjunctivitis, respiratory signs, constipation and/or diarrhea, skin hemorrhages, lethargy, and neurological symptoms like convulsions, clumsy movements, and staggering gait of infected animals. The disease in pregnant sows manifests abortions, stillbirths, mummified fetuses, and malformations [3]. CSF can cause high mortalities and morbidities in porcine populations, inflicting devastating economic losses and severely impacting the socio-economic conditions of pig farmers. The disease spreads by direct contact between pigs or through contaminated feed (swill feeding) and water, fomites, farm equipment, transport vehicle, and visitors. Moreover, the disease can also spread through infected boar semen, artificial insemination, and/or coitus. A characteristic feature of CSFV in infected pigs is noticeable immuno-suppression, including the depletion/reduction of B-lymphocytes and T-lymphocytes [4].

Disease outbreaks at regular intervals in endemic geographical regions, as well as preventive and vaccination costs to combat the virus, have great economic consequences, as the disease impacts pig breeding, national and international trade of pigs, and pork production. Because the disease has a tremendous impact on the pig industry, which ultimately affects the economy of both developed and developing countries, CSF is notifiable to the OIE, the World Organization for Animal Health [5]. Along with informing the higher authorities, there must be some standard procedures on culling methods of infected pig herds and biosecurity measures for the non-infected herds. The prohibition of trading of any pork meat products from the CSF endemic country to the disease-free region is an essential part of the strategy of control. The only effective measure to control/eradicate the disease is by way of following the vaccination strategy with cell-culture-adapted live attenuated vaccine. To control the disease, strict biosecurity guidelines and vaccination strategies have been adopted by many CSF endemic countries as part of control programs. The systematic/disciplined implementation of the vaccines, along with the coordinated control measures, could result in the eradication/elimination of the disease from both domestic and wild boar (natural reservoir) population. The CSF vaccine’s manufacturing relies on the procedures outlined in the OIE Manual of Diagnostic Tests and Vaccines for Terrestrial Animals, ensuring the production of safe and effective vaccines [6]. The Chinese (C) strain of the CSFV is a conventionally and frequently used strain for the manufacture of the vaccine globally. The C-strain-based vaccines are acclaimed highly safe and effective against the disease [7]. Recently, an indigenous CSFV isolate cell culture-based vaccine has also been developed in India. The CSF is endemic in the country, and wide variations in its prevalence are noted since its first report in 1944.

In India, the pig population is 9.06 million and shares about 1.7% of the cumulative livestock data as per the recent livestock census (20th national livestock census). Here, the pig population is mostly concentrated in the northeastern (NE) states. The average number of pigs per household is around 4.03, indicating that most of the piggery sector is maintained under a backyard condition. The number of organized farms maintaining pigs in India is less than 5000, with 100 to 2000 pigs per farm. The recent 20th Livestock Census revealed that 90.27% of the pig population is in rural India and 9.73% of pig population is reared in urban settings. The first CSF case in India was reported during 1944 in northern parts, Uttar Pradesh [8], trailed by outbreaks in West Bengal (eastern parts), and Andhra Pradesh (southern parts) during 1951 and 1959, respectively. Subsequently, a report on CSF appeared from central parts of the country, Maharashtra [9].

In this review, we intend to provide a systematic review of the prevalence and genetic diversity of the CSF virus situation in the porcine population in India, targeting complete virus genome sequence analysis, published reports on prevalence, pathology, and updates on indigenous diagnostics and vaccines from an Indian perspective.

## 2. CSFV Genome and Classification

The CSFV is a member of the genus *Pestivirus* in the family *Flaviviridae* [5]. Recently, pestivirus species are renamed and classified as Pestivirus A to K, like Bovine viral diarrhea virus (BVDV)-1 named as Pestivirus A, BVDV-2 as Pestivirus B, CSFV as Pestivirus C, and so on [10].

The CSFV genome comprises a single-stranded positive-sense RNA, of nearly 12.3 kb in length [11]. The genomic RNA is infectious because it is a positive sense and possesses a single open reading frame (ORF) with a flanked non-translated region at both the ends of the genome (5′-UTR and 3′-UTR). The ORF encodes a single polyprotein, and further downstream, processing of this polyprotein by viral and cellular enzymes generates four structural (C, E^rns^, E1, and E2) and eight/nine non-structural (N^pro^, p7, NS2-3, NS2, NS3, NS4A, NS4B, NS5A, and NS5B) proteins [12,13].

## 3. Phylogenetic and Sequence Analysis of Indian CSFV Isolates

Three genomic locations (3′end of the NS5B polymerase gene (RdRp), 5′ untranslated region (5′UTR), and E2 glycoprotein genes) are recognized to classify CSFV isolates as well as to know genetic relatedness and phylogenetic tree placements. As of now, CSFV strains are categorized in three genotypes and 3–4 subgenotypes [14,15]: (i) Genotype 1: four subgenotypes (1.1/1.2/1.3/1.4), (ii) Genotype 2: three subgenotypes (2.1/2.2/2.3), and (iii) genotype 3: four subgenotypes (3.1/3.2/3.3/3.4) [15,16]. Genotype 1 mainly contains historical strains of the virus that were retrieved globally and that contained the in-use live-attenuated vaccine strains. Genotype 2 CSFVs have been spreading since the 1980s with increasing prevalence and epidemic infections all over the world, along with two subgenotypes, namely CSFV 2.1 and 2.2, where subgenotype 2.1 is further split into 2.1a and 2.1b [17,18,19,20]. Due to the high genetic diversity among genotype 2, a few reports further suggest splitting of subgenotype 2.1 into 2.1a–2.1j [18,21]. The CSFV strains of genotype 3 are primarily found in different European and Asian (Thailand, Taiwan, Japan, Korea) regions [17]. However, all these genotypes have been reported in Asian countries [15,17].

### 3.1. CSFV Complete Genome Based Phylogenetic Analysis and Percent Similarity

We performed the phylogenetic and sequence distance analysis on 53 CSFV complete genome sequences retrieved from different Asian countries, including 14 whole-genome sequences of CSFV from India and representatives of other genotypes/subgenotypes from other countries. These sequences were retrieved from NCBI GenBank (https://www.ncbi.nlm.nih.gov/genbank/) and aligned using ClustalW in MEGA 6.0 software (Phoenix, AZ, USA) (available online: http://www.megasoftware.net/). Phylogenetic analysis was completed following the Maximum Likelihood method (1000 bootstrap replicates) [21]. The pair-wise similarity among the nucleotide sequences was calculated, and for aligning the sequences by the ClustalW program in high-speed Multiple Alignment using Fast Fourier Transform (MAFFT) online software program from the EBI website was used (https://www.ebi.ac.uk/Tools/msa/mafft/) [22].

In the phylogenetic analysis of complete genomes of the representative CSFV isolates of major genotypes, two major clades were identified, one containing genotype 1 (1.1) and the other containing genotype 2 (2.1 and 2.2) (Figure 1). Genetic divergence among the circulating Indian CSFV strains was observed, which belongs to the subgenotype 1.1, and the results are evident in their phylogenetic clustering pattern. Indian CSFV 1.1 strains were distributed in different branches inside the 1.1 clade, among which two strains (Accession no. EU857642 and MK405703) formed separate branching near Chinese isolates whereas four strains (Accession no. KM262189, MN128600, MH734359, and KC503764) from India clustered separately near a Thailand isolate. One Indian strain (Accession no. KY860615) clustered alongside South Korean strains. In subgenotype 2.2, all the Indian CSFV strains appeared distributed in several branches inside the same clade. One of the Indian CSFV strains from the state of Haryana (Accession no. MK405702) appeared to be independent inside the clade 2.2. A single strain of subgenotype 2.1 from India seemed to be divergent and appeared distantly from isolates of other countries. Apart from genotypes circulating in India, the other reported genotypes worldwide (1.2, 1.3, 1.4, 2.3, 3.2, and 3.4) constituted their respective independent clades. It is significant to note that the data set does not include two underrepresented genotypes (3.1 and 3.3) where searching of the previous literature could not reveal any accession number associated with these two subgenotypes.

Previous CSFV detection and genotyping reports from India revealed the historical prevalence of subgenotype 1.1 along with current increasing evidence and co-circulation of subgenotype 2.2 followed by 2.1 [20]. Genotype 1 contains highly virulent strains and vaccine strains, whereas genotype 2 and 3 refer to the comparatively moderately virulent strains [15]. Therefore, the changing pattern may be because of vaccine pressure [19]. The emergence of CSFV genotype 2 has already been documented in other countries like Europe, China, and Taiwan [14,17,23]. Although in India, to date, genotype 1.1 is most prevalent , analysis of genotype 2.2 has been done from a limited state only [24,25,26,27]. Therefore, the hypothesis of switching of prevalent genotypes needs further studies [26,27]. The analysis of archived sequences of Indian origin CSFV, partial or complete genome, also indicates the maximum prevalence of genotype 1.1. Notably, the phylogenetic patterns retrieved targeting the full-length (1119 bp) E2 gene of all CSFV isolates from India are similar to the whole genome-based phylogeny, suggesting that either the whole CSFV genome or E2 gene-targeted phylogeny can be used for typing and analysis of the circulating virus genotypes/subgenotypes.

As there are many Indian sequences available for CSFV 1.1 and 2.2, we have calculated the nucleotide (nt) similarity within the Indian strains (Table 1). Whole genome-based nucleotide similarity within Indian 1.1 strains was found to be between 92.075% and 96.38%, whereas it was between 83.425 and 84.99% within the whole genome of Indian 2.2 strains. The nucleotide identity of a single isolate of subgenotype 2.1 from India was 90.84%, 91.33%, 90.13%, 91.83% and 90.88% with South Korean, Mongolian, Vietnamese, Taiwanese and Japanese isolates, respectively. Whereas it ranged between 90.65% and 91.35% with Chinese isolates of 2.1 specificity.

### 3.2. Sequence Percent Homology Similarity Index of Other CSFV Gene Targets

The sequence percentage similarity of E2 and NS5B genes and 5′UTR between Indian isolates of subgenotype 1.1, 2.1, and 2.2 is given in Table 1. The percentage similarity between Indian isolates of subgenotype 1.1 is 89.2%–99.8% based on the E2 gene, 92%–99.8% based on NS5B gene, and 93.9%–98.2% based on the 5′UTR (Table 1). The higher similarity (90.7%–99.8% at nucleotide levels) of the CSFV 1.1 strains to the South Korean strain is notable as of the geographic distance, and it could possibly be due to the uses of the bovine kidney adapted CSFV vaccine imported from Korea. However, these assumptions need further detailed investigations.

Whereas, Indian 2.2 subgenotype showed similarity between 95.9% and 98.4% for E2 gene, 92.6% and 99.6% for NS5B gene, and 91% and 97.4% for 5′ UTR. As there was only a single isolate present for 2.1 subgenotype from India, its nucleotide percentage similarity was assessed with other countries 2.1 subgenotype isolates based on E2 and NS5B genes and 5′UTR. The Indian CSFV subgenotype 2.1 has shown highest similarity range of 84.2%-86.5% with Chinese CSFV strains, 89.3%–89.9% with Taiwanese, and 88.7%–95% with Chinese strains for E2 and NS5B genes, and 5′UTR, respectively.

## 4. Meta-Analysis of CSF Prevalence in India

The current meta-analysis study of CSF presents more comprehensive data compared to previous studies from India [26,27]. Additionally, CSF meta-analysis in the present study compared different diagnostic test-wise prevalence and identified the outlier and influential studies [28,29]). The basic idea to employ meta-analysis was to comprehend the scattered information on CSF disease prevalence in India over the time interval of nearly two decades. We performed a published article search to recognize all peer-reviewed articles documenting the prevalence of CSF in India using electronic databases like PubMed, ScienceDirect, Scopus, Indianjournals.com, J-Gate @Consortium of e-Resources in Agriculture (CeRA), Google Scholar, Springer, and handpicked publications (2001–2018). The keywords used for the search were CSF, India, swine, prevalence, pig, and epidemiology. All the articles on CSF prevalence in India were collected, and the Quality criteria were developed using MOOSE (Meta-analysis of Observation Studies in Epidemiology) and PRISMA (Preferred Reporting Items for Systematic Reviews) protocol. Screening at title and abstract level followed by full-text screening, data extraction, and quality assessment, were also carried out before starting the review of full papers.

All the individual studies were reviewed and screened manually by two investigators independently using both inclusion and exclusion criteria, and the third investigator resolved the discrepancy between the two investigators. The PRISMA protocol is depicted in Figure 2. The included publication was extracted into the author’s name, article title, year of publication, sample size, number of positives, study area, study year, and diagnosis method used. From the 323 papers screened (from 1980 to 2018), 23 (from 2011–2019) publications were incorporated in the systematic review and meta-analysis. The proportion for CSF prevalence was carried out using 23 studies with 79 strata level data with a total sample size of 14,123. The publication year was classified into two intervals: 2011–2015 and 2016–2019. The states which reported the prevalence of CSF were categorized into the following six regions: (i) Northern region—Jammu and Kashmir, Punjab, Uttar Pradesh, Uttarakhand; (ii) Eastern region—West Bengal, Odisha, Bihar, Jharkhand; (iii) Northeast Region—Assam, Tripura, Meghalaya, Nagaland; (iv) Western region—Rajasthan, Gujarat, Maharashtra, (v) Central region—Madhya Pradesh, Chhattisgarh; and (vi) Southern region—Kerala, Tamil Nadu, undivided Andhra Pradesh, Karnataka. The studies included in the analysis used diagnostic techniques such as AGID, ELISA, I-ELISA, S-ELISA, and RT–PCR. The details of the included studies are given in Appendix A.

Summary reports on CSF prevalence were performed by using descriptive statistics. Between-study heterogeneity was assessed graphically by visual inspection of the Baujat plot [28] and quantified by Higgin’s I^2^ and Cochran’s Q method. The meta-analysis was completed through a random effect (RE) model using the inverse-variance model [29,30]. The pooled estimate was measured and described as prevalence, with point and 95% confidence intervals (CI). Forest plots were employed to identify the prevalence in each study and the collectively estimated prevalence. Publication bias was assessed graphically by visual inspection of the funnel plot, and the Egger method [31,32]. A set of case deletion diagnostics such as studentized residuals, the difference in fits values (DFFITS), Cook’s distances, COVRATIO, and leave-one-out estimates, for the amount of heterogeneity as well as the test statistic for heterogeneity, were used to identify the influential studies [33]. The sensitivity analysis was carried out with and without the exclusion of influential studies to verify the robustness of the study design, sample size, study conclusions, and the effect of missing data. Subgroup analysis was conducted to identify the stratified prevalence in different regions, study period, diagnostic tests, and species Table 2). The R statistical platform (R Foundation for Statistical Computing, Vienna, Austria version 3.5.1 with “meta” package (version 4.9-2) and “metafor” package (version 2.0-0) was employed for statistical analyses.

From the 323 publications screened (from 1980 to 2019), 23 papers (2011–2019) were incorporated in the systematic review and meta-analysis. The other publication before 2011 reported mainly outbreaks and ambiguous samples and diagnostic tests. Hence, most of such studies were excluded from the analysis. From 23 papers, 79 strata level data were extracted. For example, the survey by NIVEDI (2008) was obtained into four strata levels representing different regions where the study was performed.

A meta-analysis of these studies showed significant variability/heterogeneity (Q = 8869.91) between the studies, and the between-study variance (Tau square) was as 0.08. The RE model revealed better symmetry than the fixed effect (FE) model and indicated that the RE model is a better one. Sub-group analysis showed a significant heterogeneity (I^2^ indices > 90%, *p*-values < 0.01) was noticed for all subgroups. In funnel plot identified publication bias (Figure 3) and due to significant publication bias (*p* = 0.41), the RE model results were considered. The Baujat plot showed that the studies that contributed to overall heterogeneity were two, and no study was identified as an influential study.

The pooled prevalence for CSF in an RE model was 35% (95% CI: 28%–43%). The sub-group analysis of diagnostic tests showed that the CSF seroprevalence with ELISA was 30% (95% CI: 22%–38%), i-ELISA 61% (95% CI: 42%–78%), s-ELISA 52% (95% CI: 0–100%), AGID 60% (95% CI: 18%–95%), RT–PCR 33% (95% CI: 8%–64%). The region-wise sub-group analysis showed CSF prevalence in the central zone (42%), East zone (41%), Northeast zone (40%), North zone (30%), southern zone (25%), and West (37%) (Figure 4; Appendix A).

The disease causes severe economic losses to pig farmers, and despite its devastating impact and recurrent outbreaks, CSF continued to be misconstrued and neglected for decades in India. The prevalence of CSF has been reported in most of the states of India [20]. In this meta-analysis, we included more data compared to the other previous studies carried out on CSF in India [26,27]. Compared to earlier studies on CSF meta-analysis in India, the present study compared the different diagnostic test-wise prevalence and identified the outlier and influential studies [28,29]. In this analysis, the prevalence estimate by a sandwich and indirect ELISA is comparatively higher than RT–PCR. The probable reason may be due to the higher false positives. In most of the studies, ELISA is often used to estimate the prevalence of CSF as it is convenient and has higher sensitivity and specificity. Serological assays provide better and quick information about CSF prevalence in the large pig population. Furthermore, serological tests are more realistic in serosurvey and extended epidemiological investigations provided assay targeted should not cross-react with other pestiviruses. The available antigen ELISA is quick but has low sensitivity [34]. As of now, the nucleic acid-based RT–PCR assay remains the method of choice due to high sensitivity for detecting virus at an early stage of infections. The possible limitations of this meta-analytic study could be that most of the studies did not clearly mention the diagnostic methodology and sampling procedures. The potential bias in the prevalence of CSF estimate might be due to low reporting and use of highly accurate RT–PCR based assays, though they are expensive.

## 5. Clinical Disease and Pathology on the CSF Outbreaks in India

CSFV is a known disease of domestic, feral, and wild suids [35]. The infection setup in vulnerable populations through the oro-nasal route and spreads via direct or indirect contact with clinically infected pigs and consuming virus-contaminated feed. Reports also support the vertical transmission of infection from sow to the offspring. The incubation period varies 3–10 days after infection. Depending on the CSFV strain, viral load, and host factors (age, breed, and immune status), CSFV infection classically takes either of the acute, chronic, or prenatal forms [36]. The clinical form of CSF has been extensively studied and presented in previous reports [36,37,38,39,40,41,42,43].

The pathological findings of CSF rely upon the clinical progression of the disease. During CSF field case investigations, we noticed a wide spectrum of clinical–pathological changes (Figure 5). In the acute course of CSF, pathology often reveals erythematous lesion in the skin of the ear, ventral surface of the abdomen, perianal region, tail, and extremities (Figure 5A). Lymph nodes (particularly mesenteric lymph nodes, inguinal lymph nodes) appear swollen with hemorrhages (Figure 5D). Serosal and mucosal surfaces of several organs such as the heart, kidneys, lungs, urinary bladder, and intestine show petechial hemorrhages. Non-collapsing, oedematous lungs with petechial to echymotic hemorrhages are frequently observed. Splenic infarctions at the edges are considered pathognomic for CSF (Figure 5B) [41]. Petecheation on the cortical surface presents a turkey egg appearance to the kidneys. Hemorrhagic enteritis (Figure 5C) and non-purulent encephalitis are also frequently observed in the acute clinical form of CSF.

The long course of the diseases in the chronic form of CSF leads to wasting in affected pigs. The acute inflammatory lesions of the initial stage of the disease are later transformed into necrotic and ulcerative lesions. Necrotic and ulcerative tonsillitis and enteritis (small intestine, colon, and ileocecal valve) are frequently observed (Figure 5E,F). The red infarction at the edges of the spleen may transform into necrotic and ulcerative lesions with the progression of the disease. Importantly, these pathological lesions vary among animals based on the host factors like age, breed and immune status, and the CSFV strain virulence [41].

Histopathological lesions of acute CSF include hemorrhagic interstitial pneumonia, hemorrhagic lymphadenitis with depletion of lymphocytes, and hemorrhagic enteritis. The alveoli and bronchiolar air spaces are often filled with sero-fibrinous exudates, necrotic debris, desquamated epithelial cells with thickened alveolar wall due to edema, congestion, hemorrhages, and mononuclear infiltration. Severe congestion and hemorrhages with depletion of lymphocytes occur in spleen and lymph nodes. Moderate to severe hemorrhages are frequently observed in the cortex and corticomedullary junction of kidneys with tubular degeneration. Similar changes but with the predominance of necrotic and ulcerative lesions were observed in chronic cases [41].

## 6. CSF Laboratory Diagnosis in India

In routine, the provisional diagnosis of CSF disease is done looking at the clinical signs and pathological changes. The laboratory confirmation of the disease is crucial to differentially diagnose it with other infectious diseases of swine [44,45,46]. Sampling a greater number of animals is advisable as this disease may progress in a chronic form [47,48]. Virus isolation in the cell culture system and subsequent characterization remains the method of choice and a gold standard as well. The virus isolation is attempted in homologous primary cells (pig kidney) or in preferred cell lines (PK-15, RK-13, SK6, PS, and swine testicular epithelioid cells). Being non-cytopathic, the CSFV growth in the cell culture system is verified by detection using several immunological techniques like immunofluorescence (FAT) or peroxidase linked assay (PLA) using staining with poly- or monoclonal antibodies.

### 6.1. Serological Methods of Diagnosis of CSF

In India, primarily serological methods are applied for surveillance epidemiological investigations. The commonly used immunological tests for detection of the virus antigen in tissue samples are ELISA (Enzyme-Linked Immunosorbent Assay) and FAT (Fluorescent Antibody Test). The antibodies in the serum samples can be detected by ELISA and VNT (virus neutralization test). Blocking ELISA with the whole virus antigen, indirect ELISA (i-ELISA), neutralization peroxidase linked assay (NPLA), complex trapping blocking ELISA, and immuno-chromatographic strip/lateral flow assays (LFA), etc. have become available in the past for the anti-CSFV antibody detection [44,45,46]. Nonetheless, these assays fail to differentiate the infected from vaccinated animals.

In India, E^rns^ and E2 proteins-based ELISAs are available for the serodiagnosis of CSFV. A double antibody-based sandwich ELISA was standardized for the detection of CSFV antigen in clinical samples [44]. Assessment of the sandwich ELISA and dot-ELISA in CSFV antigen detection in tissues of naturally infected pigs and slaughtered pigs showed 86% and 80% of the samples from diseased pigs and 20% and 14% of the samples from pigs slaughtered for human consumption positivity by the tests, respectively, and statistical analysis also showed excellent agreement between these tests [45]. A comparative evaluation of antibody-based serological assays and nucleic acid-based assays for detecting CSFV in India showed 58% to 65% positivity by sandwich ELISA and direct FAT, respectively, and 76% positivity by nested RT–PCR [46].

Recently, in India, a recombinant Newcastle disease virus (NDV) viral expression vector was developed expressing the CSFV E2 and E^rns^ proteins and inducing production of CSFV-neutralizing antibodies on pig inoculation. Its diagnostic potential was assessed in an indirect ELISA measuring antibody titers in serum samples [47]. In another study, Bhattacharya et al. [48], employing a lentivirus-based gene delivery system, constructed a stable PK-15 cell line expressing E^rns^ (PK-E^rns^) for using it to develop an ELISA detecting E^rns^-specific antibodies in pig sera helping in differentiation of infection from vaccinated animals. Notably, a study from the northeastern region of India reported detection of CSFV in bovine samples after screening 134 cattle serum samples using a commercial antigen capture ELISA, where 10 samples were found positive for CSFV antigen by ELISA [49].

### 6.2. Molecular Methods of Diagnosis of CSF

In the current era, molecular tools are considered the backbone of disease diagnosis, genotyping, and analysis of virulence of the virus [50]. Among the several molecular tools available, real-time PCR (quantitative-PCR or qPCR) appears as the method with the highest sensitivity for the CSFV detection [51,52] and has been adopted as a test of choice for confirmatory diagnosis of the disease. Using the real-time PCR assay, CSFV was detected in tissues from 1120 slaughtered pigs in India, providing baseline prevalence data on CSF infection [40]. For CSFV RNA detection in infected tissue, a fluorescence-based in-situ hybridization (FISH) based method has been reported. The FISH assay uses a biotinylated DNA probe targeting the E2/NS2 gene of CSFV. This technique helped in demonstrating CSFV nucleic acids in the lymphoid tissues, such as spleen and lymph nodes [53]. The RT–PCR assays have been employed to detect CSFV RNA in formalin-fixed tissues, making it useful where the supply of fresh biological samples is difficult [54].

Immunochromatographic strip assay using a poly- or monoclonal colloidal gold conjugation system is being in use as a rapid pen-side test for the detection of CSF virus antigen [55]. Further, immunomagnetic bead-based assay use in the detection of CSFV antigen has been reported [56]. Another useful assay is the RT-loop-mediated isothermal amplification (RT-LAMP), which is a highly rapid 100-fold more sensitive than conventional gel-based RT–PCR while detecting CSFV. Further, it was highly specific and could differentiate other viruses like BVDV, porcine reproductive and respiratory syndrome virus (PRRSV), swine influenza virus (SIV), porcine parvovirus (PPV), porcine circovirus (PCV), and pseudorabies virus (PRV). Several advantages, like low-cost input (devoid of any specific instrument) and quick results, makes it an excellent assay for CSFV surveillance in the field. The development of qPCR and multiplex qPCR for the diagnosis of CSFV and simultaneous detection of CSFV has also been reported [57,58].

### 6.3. Compelete Genome Sequencing of Indian CSFV Isolates

Recently, different research groups from India have provided a complete genome sequence of local isolates from India using either next-generation sequencing methodology or conventional RT–PCR method. In a study, targeting the overlapping fragments of CSFV in RT–PCR, the complete genome of a lapinized CSFV vaccine strain was retrieved. The genetic analysis showed 92.6%–98.6% similarities at the nucleotide level with other CSFV strains, and it was typed as subgroup 1.1. The 5’-UTR had more than 97.0% similarity with several CSFV vaccine strains from China [59]. Subsequently, the first whole genome of a CSFV subgroup 2.2 (CSFV/IND/UK/LAL-290) was reported from the Uttarakhand state of India recovered from a backyard pig [60]. At the same time, a complete genome from a CSFV field isolate of subgenotype 1.1 was reported from India [61]. Subsequently, a complete genome from CSFV subgroup 2.1 that caused local outbreak in the northeastern state, Assam, India was sequenced. The isolate exhibited high genetic divergence [62]. Another CSFV genotype 1.1 isolate adapted in a porcine kidney cell line was deciphered with some T insertions in 3′ UTR [63]. The same group successively provided a complete genome sequence of CSFV strain (CSFV-UP-BR-KHG-06), genotype 2.2 [64].

## 7. CSF Virus Vaccines in India

Keeping the prioritization for the CSF control in the country, the Ministry of Fisheries, Animal Husbandry and Dairying, Govt. of India has initiated the Classical Swine Fever Control Programme (CSF-CP) in the year 2014–2015 to control the CSF in pigs by mass vaccination using the live attenuated vaccines. There are two major activities in this control program: (i) strengthening of laboratories, including consumables for laboratories; and (ii) vaccination in identified villages including vaccination cost, on a 90:10 allocation basis between centre and the northeastern states. The variation in quality, safety, efficacy, and potency of CSF live vaccines are major hindrances in the success of vaccination programs, and it is, therefore, a significant step to perform thorough quality control of vaccines from different suppliers regularly/batch-wise.

The CSF vaccines are to be produced following the standard operating procedures listed in the OIE Manual of Diagnostic Tests and Vaccines for Terrestrial Animals to ensure a high level of efficacy and safety [6]. The systematic prophylactic vaccination and a stamping-out policy is followed usually to restrain CSF in endemic countries. In India, CSF is endemic, and for its control, a mass vaccination approach is adopted. The lapinized vaccine using the Weybridge strain of the virus, which belongs to the sub-genogroup 1.1, has been used since 1964 [2,7]. The lapinized vaccines are produced mostly by the Institute of Veterinary Biologicals located in different states of the country to meet the local demand of the vaccines. A few of the Institute of Veterinary Biologicals have lately shifted to producing local CSFV strain based cell culture attenuated live vaccines [65]. The systematic vaccination using a live-attenuated vaccine appears to be the best way forward for elimination/eradication of the CSF, including vaccination of the reservoir hosts like wild boars [66].

The present domestic pig population of India is 9.06 million, as per the 20th livestock census of the government of India. The vaccination coverage of all the domestic pigs twice in a year requires about 20 million doses per annum, and only 1.2 million doses are produced per year by the lapinized vaccine. The reason behind this is that only 50 doses of vaccines are produced from a single rabbit spleen. To meet out the demand and overcome the constraints in producing a large quantity of the lapinized vaccine, attempts are underway to produce a cell-culture-based vaccine using either the lapinized vaccine strain or the local CSFV isolates. There are reports on developing effective cell-culture-based live attenuated vaccines against CSFV (using foreign strain, Weybridge) by the Indian Veterinary Research Institute (IVRI), the premier veterinary institute of the country. The vaccine was reported to be safe, potent, and provides immunity for a period of one year [63]. A commercial Bovine Kidney cell culture adapted vaccine for CSFV (Himmvac Hog Cholera (T/C) Vaccine) is also available from KBNP Inc. Korea through Panav Bio-Tech, India.

On 3 February 2020, IVRI released a new safe, potent live-attenuated CSFV cell culture vaccine using indigenous strain (Press Information Bureau, 2020). The vaccine would be the best choice for use in the CSF Control Programme (CSF-CP) already launched by DAHD, Govt. of India. There is a huge demand for transfer of this vaccine technology from various state governments and private manufacturers, and the vaccine has huge export potential, especially to Asian countries. Due to a very high titer of vaccine virus, this vaccine would be the most economical CSF vaccine, costing around less than Rs 2/- per dose, compared to Rs 15-25/- of lapinized CSF vaccine and Rs. 30/dose (approx.) for an imported Korean vaccine being used in the country. Besides, the new vaccine gives immunity for two years as compared to 3 to 6 month’s protection under the vaccines currently being used. The vaccine is safe, potent, does not revert to virulence, and provides protective immunity from 14-day post-vaccination (DPV) of the vaccination until 2 years, studied so far. The yield of this vaccine is 1000 times more than the existing CSF vaccine. The evaluation of different parameters along with the protective humoral immune response of pigs vaccinated with a live attenuated cell culture vaccine has been reported from the state of Assam, India. The vaccine was reported to be safe, elicited better and stable immune response after booster doses, and maternally derived antibodies persist up to 42 days in newborn piglets when pregnant pigs are vaccinated at one month of gestation [66].

## 8. Conclusions and Prospects

CSF is one of the dreaded diseases affecting mostly pigs and wild boars worldwide, causing a severe impact on the global economy. In India, the disease is endemic and mainly present in the northeastern states of the country and pig-populated states. The genotypes 1.1, 2.1, and 2.2 are circulating all over India, with the highest prevalence of genotype 1.1. Recently, increases in prevalence of 2.1 and 2.2 are also reported but need exhaustive epidemiological studies before deriving the conclusion. In India, several socio-economic impacts pose challenges to researchers for the development of potent vaccines against CSFV. Backyard pig farming and highly mobile pig products are the major factors regarding the epidemiology of CSFV in India. The key factors of the CSFV control program should include better diagnostic facilities, potent and effective vaccination, and stamping out along with control of animal movements and biosecurity. Regarding diagnostics, there is a need for rapid and cost-effective point-of-care assay to screen the disease at the farmer’s level. Along with CSFV infection, secondary infections are widespread, which sometimes overshade the disease manifestation. The multiplex assay will be better in these situations to detect concurrent infections. To avoid the occasionally observed cross-reaction with pestiviruses, more reliable and accurate CSFV-specific screening assays must be employed for confirmation. Although many ELISAs are available detecting CSFV specifically, the definitive confirmation must use serum neutralization test. Regarding vaccination strategy, the marker vaccines should be chosen for effective vaccination, which can differentiate infected and vaccinated animals. Finally, the culling of infected animals and the adoption of animal identification systems to trace the disease transmission pathway will be helpful in disease control. To eradicate the disease in near future, there is a need for increasing awareness about CSFV, mainly in the poor population who are mostly engaged with pig rearing.

## Figures and Tables

**Figure 1 pathogens-09-00500-f001:**
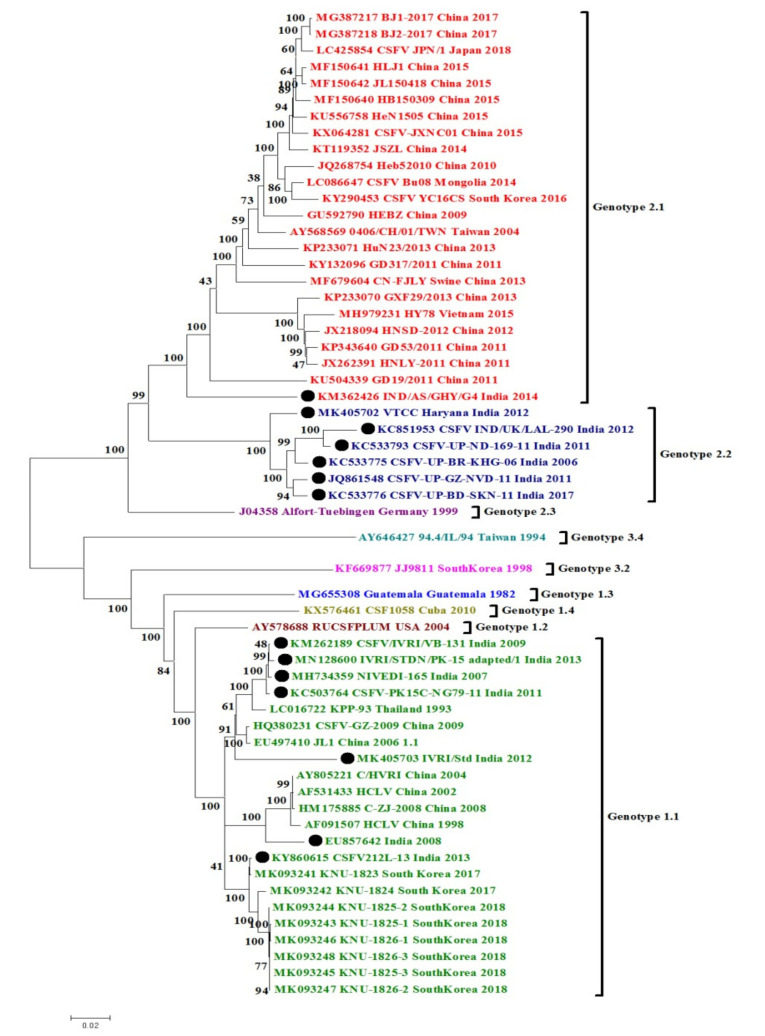
Phylogenetic analysis of Asian classical swine fever virus (CSFV) strains based on the complete genome. The different genotypes used in the current study are depicted in different color codes, and Indian CSFV strains are designated in solid black dots. Phylogenetic analysis was achieved following the Maximum Likelihood method (1000 bootstrap replicates) based on the General Time Reversible model in MEGA 6 software (v 6.06, Phoenix, AZ, USA).

**Figure 2 pathogens-09-00500-f002:**
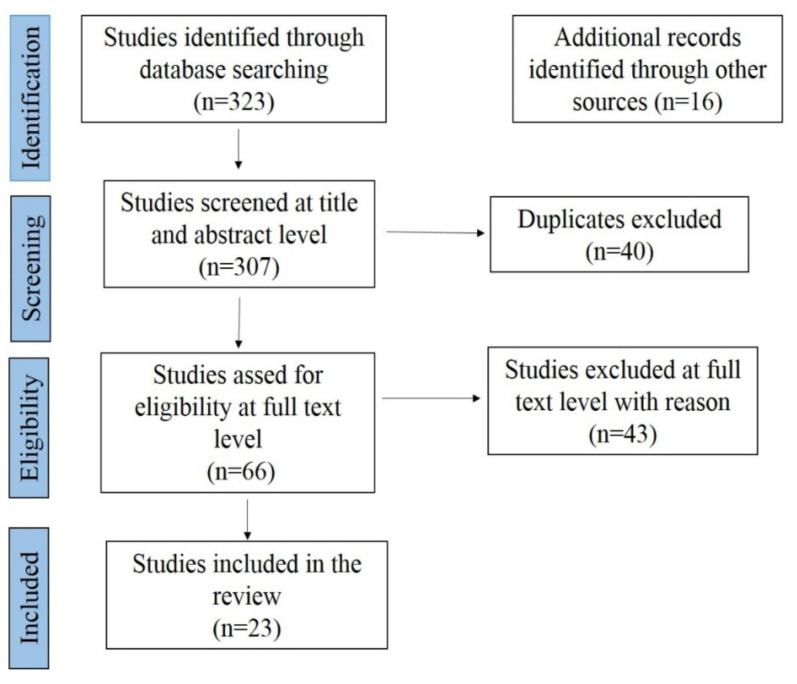
Schematic diagram showing the PRISMA (Preferred Reporting Items for Systematic Reviews) chart for the studies from India on classical swine fever (CSF) prevalence.

**Figure 3 pathogens-09-00500-f003:**
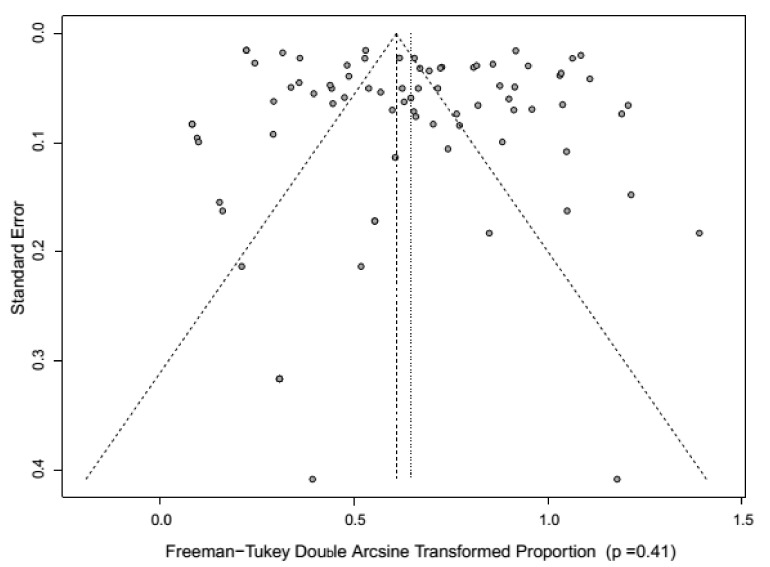
Funnel plot of the included articles for analysis demonstrates potential publication bias.

**Figure 4 pathogens-09-00500-f004:**
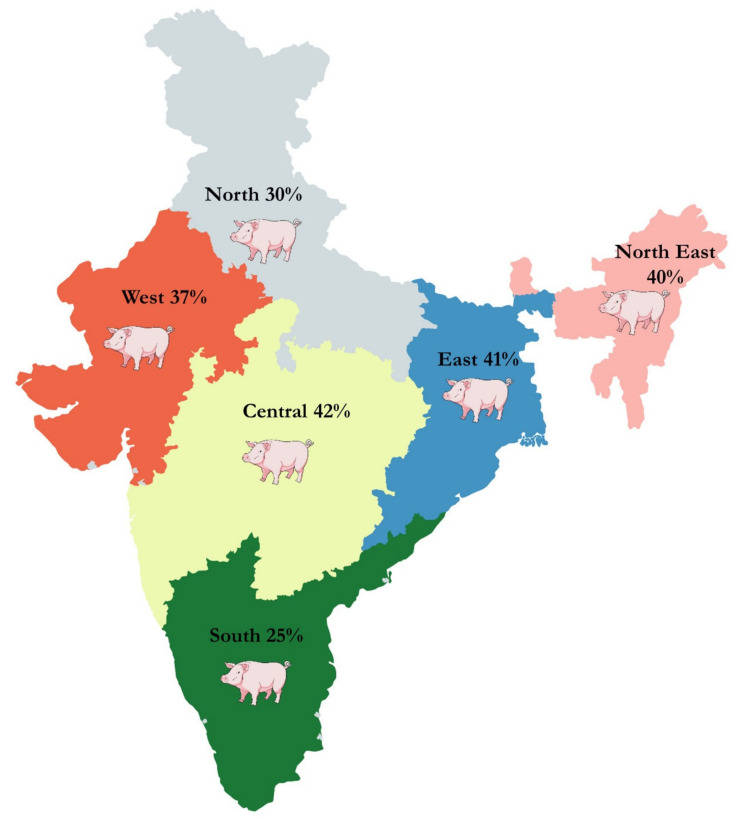
The depiction of region-wise CSF percentage prevalence in central, eastern, northeastern, northern, southern, western, and central zones of India.

**Figure 5 pathogens-09-00500-f005:**
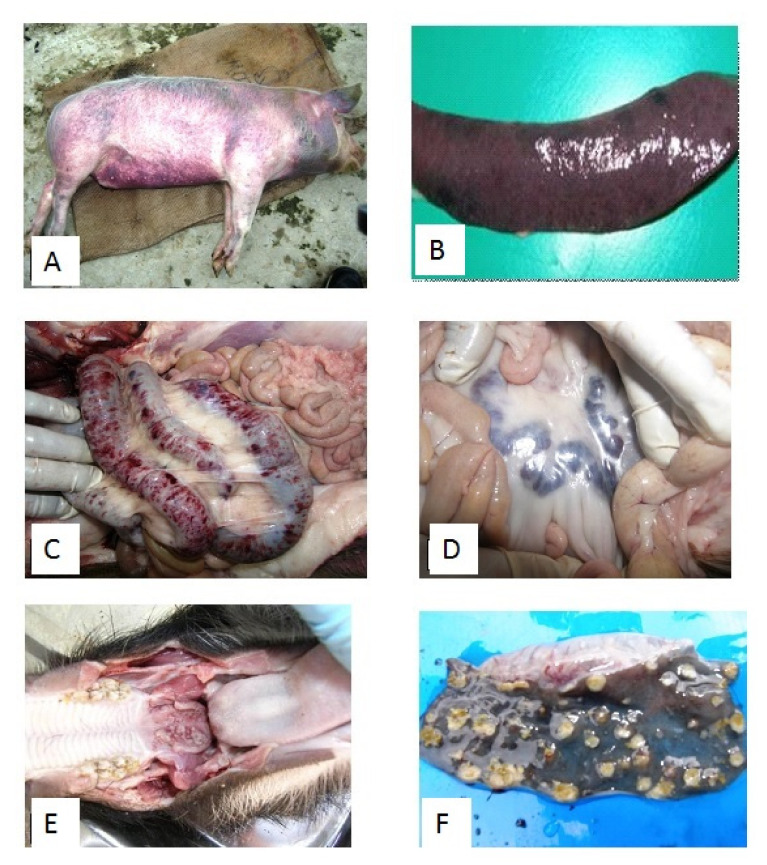
Acute and chronic form of CSF related lesions observed during field investigations in India. (**A**), Hemorrhages in skin, acute CSF form; (**B**), infarction in spleen, acute CSF form; (**C**), Hemorrhagic enteritis, acute CSF form; (**D**), hemorrhagic mesenteric lymph nodes, acute CSF form; (**E**), necrotic tonsillitis, chronic CSF form; and (**F**), Button ulcers in colon, chronic CSF form.

**Table 1 pathogens-09-00500-t001:** Nucleotide percent similarity based on whole genome, E2, NS5B genes, and 5′UTR of Indian CSFV strains with isolates from different Asian countries.

CSFV Genotype	Gene/Target	Country	Nt Similarity (%) India
**CSFV 1.1**	**Whole Genome**	India	92.07–96.38
		China	94.8–98.16
		South Korea	93.15–99.88
	**E2**	India	89.2–99.8
		China	90.1–97.5
		South Korea	90.7–99.8
	**NS5B**	India	92–99.8
		China	92.1–97.6
		South Korea	93.1–99.1
	**5′ UTR**	India	93.9–98.2
		China	91–96.8
		South Korea	91.6–98.4
**CSFV 2.2**	**Whole Genome**	India	83.42–84.99
	**E2**	India	95.9–98.4
	**NS5B**	India	92.6–99.6
	**5′ UTR**	India	91–97.4
**CSFV 2.1**	**Whole Genome**	South Korea	90.84
		China	90.65–91.35
		Mongolia	91.33
		Vietnam	90.13
		Taiwan	91.83
		Japan	90.88
	**E2**	South Korea	84.2–85.4
		China	84.2–86.5
		Mongolia	84.4–85.7
		Vietnam	83.7––84.8
		Taiwan	85.1–86.3
		Japan	84.3–85.6
	**NS5B**	South Korea	88.9–89.1
		China	88.5–89.7
		Mongolia	89–89.3
		Vietnam	87.9–88.1
		Taiwan	89.3–89.9
		Japan	88.7–89.2
	**5′ UTR**	South Korea	90–93.4
		China	88.7–95
		Mongolia	89.4–92.6
		Vietnam	89.4–91.8
		Taiwan	90.8–94.2
		Japan	62–63.1

**Table 2 pathogens-09-00500-t002:** Details of the sub-group analysis for seroprevalence of CSF in India.

S. No	Variables		Samples Tested	Positive Samples	Pooled Estimate (RE) (95% CI)	Pooled Estimate (FE) (95% CI)	*p*-Value	I^2^ Value	Tau Square
1.	Geographic region	Northern India	2569	272	30% (14%–50%)	8% (7%–9%)	<0.01	99%	0.05
		Western India	332	145	37% (8%–73%)	39% (34%–45%)	<0.01	98%	0.22
		Central India	593	302	42% (24%–61%)	51% (47%–55%)	<0.01	94%	0.04
		Southern India	3661	883	25% (18%–33%)	22% (21%–23%)	<0.01	95%	0.03
		Eastern India	54	28	41% (10%–75%)	52% (37%–66%)	<0.01	80%	0.09
		North Eastern India	6064	2678	40% (29%–51%)	43% (41%–44%)	<0.01	98%	0.07
		India	1207	786	72% (54%–87%)	65% (63%–68%)	<0.01	95%	0.01
2.	Serological test	ELISA	9224	3200	30% (22%–38%)	31% (30%–32%)	0.00	98%	0.08
		RT–PCR	1988	423	33% (8%–64%)	17% (15%–18%)	< 0.01	99%	0.10
		S-ELISA	357	27	52% (0%–100%)	4% (2%–7%)	< 0.01	97%	0.63
		AGID	196	136	60% (18%–95%)	71% (64%–77%)	<0.01	97%	0.09
		I-ELISA	2605	1253	61% (42%–78%)	48% (46%–50%)	<0.01	99%	0.07
		IIP	110	65	59% (50%–68%)	59% (50%–68%)	NA	NA	NA
3.	Study period	2011-15	9019	2356	36% (28%–43%)	21% (20%–22%)	0.00	98%	0.07
		2016-19	5461	2738	35% (24%–47%)	50% (48%–51%)	<0.01	98%	0.06

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
