# Peer review of "Classical Swine Fever Virus Biology, Clinicopathology, Diagnosis, Vaccines and a Meta-Analysis of Prevalence: A Review from the Indian Perspective"

_pathogens, 2020, doi:10.3390/pathogens9060500_

Round 1

Reviewer 1 Report

This is a review format submission paper to describe the biology and epidemic status of classical swine fever (CSF) in India. This paper will help us to understand the biological features and prevalence of CSF in India and may be of benefit to the readers. However, the reliability of the data in the paper and the immaturity of the text as a review format are significant. Hence, this reviewer judges that the adoption of this paper requires substantial revision.

1. The review should explain the biology of the virus isolates and the epidemiology of CSF in India; the basic information on CSF virus (CSFV) given in P2-P4 should be deleted.

2. The genetic characteristics of CSFVs isolated in India can be fully understood in Fig. 3; suggesting that Figs. 4 and 5, which compare the partial genes, should be deleted. Or only the most common E2 phylogenetic tree (i.e., Fig. 4) should be shown.

3. Reference virus strains of all genotypes should be included in the phylogenetic tree. If we look at this phylogenetic tree, we will easily misunderstand that there are only three genotypes (1.1, 2.1, and 2.2) in CSFVs.

4. A meta-analysis was conducted to determine the prevalence of the disease. Is there a 30% “prevalence” of CSF in India? Or are these numbers shown the “seroprevalence”?

5. If the “prevalence” of CSF in India is 30%, how much vaccination is being administered at present? And if “seroprevalence” is set at 30%, what is the actual “prevalence” rate based on your analysis?

6. The information on pathology and diagnosis on pages 14-17 is also unnecessary for this review and should be deleted.

Author Response

S. No

Reviewer’s comment

Response

Reviewer 1

1.

The review should explain the biology of the virus isolates and the epidemiology of CSF in India; the basic information on CSF virus (CSFV) given in P2-P4 should be deleted.

We have tried to reduce the basic information on CSFV.

2.

The genetic characteristics of CSFVs isolated in India can be fully understood in Fig. 3; suggesting that Figs. 4 and 5, which compare the partial genes, should be deleted. Or only the most common E2 phylogenetic tree (i.e., Fig. 4) should be shown.

As suggested, Figures of E2, NSP5 and 5UTR have been deleted. One phyloanalysis figure is kept now based on complete genome. 

3.

Reference virus strains of all genotypes should be included in the phylogenetic tree. If we look at this phylogenetic tree, we will easily misunderstand that there are only three genotypes (1.1, 2.1, and 2.2) in CSFVs.

The phylogenetic analysis was reconstructed taking all the virus strains from genotypes 1, 2 and 3 their subgenotypes 1.1, 1.2, 1.3, 1.4, 2.1, 2.2, 2.3, 3.1, 3.2, 3.3, and 3.4 to give a comprehensive view of other genotypes also.  

4.

A meta-analysis was conducted to determine the prevalence of the disease. Is there a 30% “prevalence” of CSF in India? Or are these numbers shown the “seroprevalence”?

Its 35 % prevalence including both antibody and antigen prevalence

5.

If the “prevalence” of CSF in India is 30%, how much vaccination is being administered at present? And if “seroprevalence” is set at 30%, what is the actual “prevalence” rate based on your analysis?

The details of CSF vaccination in India is not clear. The vaccination is mainly performed in organized farms not in field conditions.

Only few studies used RT-PCR for antigen identification. If we take seroprevalence and the PCR prevalence the CSF prevalence rate in this study is 35%. By RT-PCR the prevalence is 33%. This has now been added in manuscript as well

6.

The information on pathology and diagnosis on pages 14-17 is also unnecessary for this review and should be deleted.

The purpose of providing the pathology section was to detail the findings on field investigations with typical signs. The work was carried out in author’s laboratory. The title of the manuscript has been modified in view of another reviewer’s suggestions and that accommodates this section also. 

Reviewer 2 Report

The manuscript is well written and cover important aspects of the disease. However, certain parts of the document should be revised and the title does not reflect correctly the content of the article and the conclusions; in the title should be mentioned that is a “review”.

It is not clear why details on lab diagnosis, the disease, pathology, etc. should be relevant for the article which, based on the title is focused on the biology of the VIRUS and the meta-analysis of prevalence

There are inferences about disease control strategies but it is not the subject of the paper and in case they should be better substantiated by data.

  • Line 74: ..is an essential part of the strategy of control. It should replace biosecurity

  • Line 141: In UK, CSF was eradicated in 2020.
  • Line 249: delete “,”
  • Line 312 – 317: from what is reported in the paragraph it is not clear if a differential diagnosis from others pestiviruses (by serology) is performed
  • Lines 488 – 489: In Europe in the last 15 years, domestic pigs have never been vaccinated against CSF.
  • Lines 488 – 496: updated references should be included to substantiate the paragraph.
  • Line 529: affecting mostly young pigs and wild boar. “Young” should be deleted, also adult pigs are “affected”, they might no show clinical signs but this is different.
  • Lines 535-536: is there a reference to quote?
  • Line 538: for the control of ASF biosecurity is also essential
  • Line 542: In relation to the serological cross reactivity with other pestivirus, please see the OIE diagnostic manual and address the text accordingly, in case you consider this part still relevant for you manuscript: “As CSFV cross-reactive antibodies against other pestiviruses are occasionally observed in pigs, screening tests have to be followed by confirmatory tests that are CSFV-specific. Certain ELISAs are relatively CSFV-specific, but the definitive method of choice for differentiation is the comparative neutralisation test, which compares the neutralising titre of antibodies to different Pestivirus isolates

Comments on the statistical methods

Statistical methods are well described in the text, however some points need clarification:

  1. a table with the list of the studies included in the review is essential. For each paper the type of study design, number of events, number of animals evaluated, prevalence and 95%CI, geographic region, serologiacal test and study period should be reported;
  2. p-value of the funnel plot should be provided;
  3. line 257: modify RE in random effect (RE);
  4. meta-analysis: If the aim of the analysis is to evaluate the effect of the serological techniques, the studies should not be splitted also by geographical region because this would cause a decrease of the power of the statistical results. The same apply to the others factors considered in the analysis.
  5. Supplementary material: Certain parts of the plot are written in grey. They tests are not well visible and figures are flattened, the quality of images needs improvement;
  6. To be aligned with the text of the manuscript, the numbers of the figures of the graph named “forest plot” should be switched (1 into 2 and vice versa).

Author Response

S. No

Reviewer’s comment

Reply

Reviewer 2

1.

Certain parts of the document should be revised and the title does not reflect correctly the content of the article and the conclusions; in the title should be mentioned that is a “review”.

Few of the sections have been modified and title of the review has been changed to reflect the contents of the manuscript correctly

2.

It is not clear why details on lab diagnosis, the disease, pathology, etc. should be relevant for the article which, based on the title is focused on the biology of the VIRUS and the meta-analysis of prevalence

The purpose of the review was to give an overview of CSFV research in India including analysis of circulating strains and meta-analysis of prevalence. To fit these sections the title of review has been modified. 

3.

There are inferences about disease control strategies but it is not the subject of the paper and in case they should be better substantiated by data.

The sentences have been re-phrased.

·         Line 74: …is an essential part of the strategy of control. It should replace biosecurity

Changed as suggested

·         Line 141: In UK, CSF was eradicated in 2020.

Corrected

·         Line 249: delete “,”

It has been deleted

·         Line 312 – 317: from what is reported in the paragraph it is not clear if a differential diagnosis from others pestiviruses (by serology) is performed

The sentences have been re-phrased.

·         Lines 488 – 489: In Europe in the last 15 years, domestic pigs have never been vaccinated against CSF.

·         Lines 488 – 496: updated references should be included to substantiate the paragraph.

Done as per the suggestions

·         Line 529: affecting mostly young pigs and wild boar. “Young” should be deleted, also adult pigs are “affected”, they might not show clinical signs but this is different.

Corrected as per the suggestion

·         Lines 535-536: is there a reference to quote?

This is the observation of the authors of this review. They are working in north-eastern states of the country having maximum number of pig population and cases of CFV also

·         Line 538: for the control of CSF biosecurity is also essential

We agree, it is not inserted as per the suggestion  

·         Line 542: In relation to the serological cross reactivity with other pestivirus, please see the OIE diagnostic manual and address the text accordingly, in case you consider this part still relevant for you manuscript: “As CSFV cross-reactive antibodies against other pestiviruses are occasionally observed in pigs, screening tests have to be followed by confirmatory tests that are CSFV-specific. Certain ELISAs are relatively CSFV-specific, but the definitive method of choice for differentiation is the comparative neutralisation test, which compares the neutralising titre of antibodies to different Pestivirus isolates

Sentence re-phrased accordingly.

Comments on the statistical methods

Statistical methods are well described in the text, however some points need clarification:

1

A table with the list of the studies included in the review is essential. For each paper the type of study design, number of events, number of animals evaluated, prevalence and 95%CI, geographic region, serological test and study period should be reported;

The list of the studies included in the meta-analysis   are given in supplementary file

2

p-value of the funnel plot should be provided;

Provided in the funnel plot

3

line 257: modify RE in random effect (RE);

Modified in the revised MS

4

Meta-analysis: If the aim of the analysis is to evaluate the effect of the serological techniques, the studies should not be splited also by geographical region because this would cause a decrease of the power of the statistical results. The same apply to the others factors considered in the analysis.

In region and year wise subgroup analysis there is enough observations in each category the statistical power has not decreased.

5

Supplementary material: Certain parts of the plot are written in grey. They tests are not well visible and figures are flattened, the quality of images needs improvement;

Improved the image quality and changed the grey parts.

6

To be aligned with the text of the manuscript, the numbers of the figures of the graph named “forest plot” should be switched (1 into 2 and vice versa).

Done as per suggestion

Round 2

Reviewer 1 Report

The contents of this paper have been relatively improved in response to comments on the first submission. However, there is too much description of general information about CSF and CSFV. Efforts should be made to convey the main idea of the paper - the status of CSF in India - to the readers concisely.

<Major points>

  1. Line 42: As I already commented in your first submission, the prevalence (35.4%) calculated here should be carefully explained as the total number based on data from both antigen and antibody tests.
  2. This is a review paper of CSF and CSFV in India. Therefore, basic information on general virology, pathology, and diagnosis should be kept to a minimum.

2-1 Line 104-133: The description of this information should be kept to a minimum. The information in this paragraph should also be included in the section on genetic analysis of the isolated virus, beginning with Line 140.

2-2 Line 323-342: This information should be kept to a minimum.

2-3 Line 376-394: This description of information should be kept to a minimum.

2-4 Line 396-430: This description of information should be kept to a minimum.

  1. Figure 1 should be deleted as I suggested in #-2-1.
  2. Line 205-209: ???
  3. Line 140: Title: Phylogenetic and sequence analysis of CSFV isolates in India
  4. Line 322: Title: Clinical disease and pathology on the CSF outbreaks in India
  5. line 375: Title: CSF Laboratory Diagnosis in India

<Minor points>

  1. Line 2: “(CSF)” should be removed from the title.
  2. Line 68: CSFV
  3. The branch numbers in Figure 6 should be a, b, and c, not 1, 2, and 3.

11. Ernsand Npro should be superscript.

Author Response

Dear Sir/Madam

I submit herewith the response to all your suggestions. I have tried to reduce the general information. 

Suggestion: The contents of this paper have been relatively improved in response to comments on the first submission. However, there is too much description of general information about CSF and CSFV. Efforts should be made to convey the main idea of the paper - the status of CSF in India - to the readers concisely.

Response: The revised version is modified to reduce the general information and focus is now on the status of the CSF/CSFV in India.

Major points

Suggestion 1: Line 42: As I already commented in your first submission, the prevalence (35.4%) calculated here should be carefully explained as the total number based on data from both antigen and antibody tests.

Response: Modified as per the suggestion.

Suggestion 2: This is a review paper of CSF and CSFV in India. Therefore, basic information on general virology, pathology, and diagnosis should be kept to a minimum.

Response: The revised version is modified to reduce the general information and focus is now on the status of the CSF/CSFV in India.

2-1 Line 104-133: The description of this information should be kept to a minimum. The information in this paragraph should also be included in the section on genetic analysis of the isolated virus, beginning with Line 140.

Response: The revised version is modified to reduce the description.

2-2 Line 323-342: This information should be kept to a minimum.

Response: The revised version is modified to reduce the description.

2-3 Line 376-394: This description of information should be kept to a minimum.

Response: The revised version is modified to reduce the description.

2-4 Line 396-430: This description of information should be kept to a minimum.

Response: The revised version is modified to reduce the description.

  1. Figure 1 should be deleted as I suggested in #-2-1.

Response: Figure 1 deleted in the revised version.

  1. Line 205-209: ???

Response: The purpose is to give a sequence analysis of other genes that are used for the genotyping of CSFV. This section has been re-phrased to clear the message in the revised version.

  1. Line 140: Title: Phylogenetic and sequence analysis of CSFV isolates in India

Response: Revised in the new version as per suggestion.

  1. Line 322: Title: Clinical disease and pathology on the CSF outbreaks in India

Response: Revised in the new version as per suggestion.

  1. line 375: Title: CSF Laboratory Diagnosis in India

Response: Revised in the new version as per suggestion.

Minor points

  1. Line 2: “(CSF)” should be removed from the title.

Response: Revised in the new version as per suggestion.

  1. Line 68: CSFV

Response: Revised in new version as per suggestion.

  1. The branch numbers in Figure 6 should be a, b, and c, not 1, 2, and 3.

Response: Revised in the new version as per suggestion.

  1. Erns and Nproshould be superscript.

Response: Revised in the new version as per suggestion.

Round 3

Reviewer 1 Report

The paper has been corrected according to this reviewer's suggestions. Please correct the following three small points.

  1. Line 42; delete " ) ".
  2. Line 312; If you will use 5.1, where is 5.2?
  3. Line 331; "Erns" ???

Author Response

The paper has been corrected according to this reviewer's suggestions. Please correct the following three small points.

Response: Thanks for considering our revision.

Suggestion 1: Line 42; delete " ) ".

Response: Delete “)”

Suggestion 2: Line 312; If you will use 5.1, where is 5.2?

Response: Subheading 5.1 deleted

Suggestion 3: Line 331; "Erns" ???

Response: Couldn’t locate this correction at 331 line or next to this line also. Please see in the clean file attached.
